# OpenReview forum: "On the Efficiency-Safety Dilemma in Large Reasoning Models"
_ICLR.cc/2026/Conference — Submitted to ICLR 2026_

### Official Review · Reviewer_RM16 · 2025-10-29

**Soundness:** 3
**Presentation:** 3
**Contribution:** 2
**Rating:** 4
**Confidence:** 4

**Summary:**

The paper introduces the first systematic benchmark analysis of the influence of compression techniques on efficiency (accuracy and runtime/resources) and safety applied to Reasoning Models. The authors compared 3 reasoning models using 6 compression techniques (2 implementations of 3 methods - quantization, pruning, and KV Cache compression). The paper found that some compression techniques enabled better trade-off between accuracy and enhanced safety. However, the author's claimed that enhanced safety often comes from degraded performance (correlation and behaviour analysis).

**Strengths:**

Efficiency methods often counter-intuitively improve the safety. Well shown given the experimental settings, and we can see that performance can be largely retained while enhancing robustness.

Robustness of reasoning models is competitively under-explored, thus more research towards this topic is a timely considering the increasing prominence of LRMs.

**Weaknesses:**

A lot of papers have been published in the area of understanding the impact of compression techniques toward safety (e.g. [1, 2, 3]). This paper does not link its findings to classic LLMs. It does not not clearly assess the influence of the reasoning component - are there new fundamental insights into this study compared to prior works or do LRM behave as expected from earlier research? How would the claims differ in on classic LLMs?

[1] Kumar, Divyanshu, et al. "Fine-tuning, quantization, and llms: Navigating unintended outcomes." arXiv preprint arXiv:2404.04392 (2024).

[2] Belkhiter, Yannis, Giulio Zizzo, and Sergio Maffeis. "Harmlevelbench: Evaluating harm-level compliance and the impact of quantization on model alignment." arXiv preprint arXiv:2411.06835 (2024).

[3] Kharinaev, Artyom, et al. "Investigating the impact of quantization methods on the safety and reliability of large language models." arXiv preprint arXiv:2502.15799 (2025).

The paper is styled as a benchmarking and empirically driven paper, i.e. there is no theoretical analysis carried out why some methods affect safety more than others. However, the sample sizes for an empirical evaluation seem limited: Evaluating over 100 (50 for each) core sentences for jailbreaks is small for a benchmarking evaluation. Further, two of the models are distilled from DeepSeekR1, potentially introducing correlation between the results seen for R1D-Qwen-32B and R1D-Llama-8B.

**Questions:**

The H-COT dataset release on Huggingface contains pre-constructed strings, so seems like it it a non-adaptive. Likewise the Mousetrap paper has a HF dataset of the attack output. Are the results presented here therefore against non-adaptive transfer attacks against unquantized models?

---

### Official Review · Reviewer_VyC2 · 2025-10-30

**Soundness:** 2
**Presentation:** 2
**Contribution:** 3
**Rating:** 4
**Confidence:** 3

**Summary:**

This paper provides the first comprehensive analysis of how efficiency techniques, such as quantization, pruning, and KV Cache compression, affect the safety of Large Reasoning Models (LRMs). The study finds that while most efficiency methods seem to improve safety against jailbreak attacks, this improvement is superficial. This apparent safety gain is not due to better alignment but is a side effect of reduced reasoning ability, which causes the models to produce more "attempted but failed" malicious responses. The research confirms a consistent correlation, finding that higher reasoning performance is linked to greater vulnerability to jailbreak attacks. Ultimately, the paper identifies a combination of quantization with pruning as the optimal strategy to best balance efficiency, safety, and reasoning performance.

**Strengths:**

1. The study provides the first comprehensive analysis of how common efficiency techniques (like quantization, pruning, and KV Cache compression) impact the safety of Large Reasoning Models (LRMs). This is an important problem in practice, given the widespread use of these efficiency methods in real-world deployment.
2. The authors use a relatively robust experimental setup, with multiple benchmarks, LRMs, and efficiency methods. Beyond surface-level results, like the finding that efficiency methods appear to improve safety, the analysis digs deeper to find the true cause: "degraded reasoning capacity", leading to more "attempted but failed answers" rather than genuine alignment.
4. The study goes beyond individual techniques to evaluate combinations, which reflect realistic deployment scenarios. This allows the paper to provide an empirical recommendation: quantization combined with pruning is identified as the "optimal strategy" for best balancing efficiency, safety, and reasoning performance, offering a clear takeaway for practitioners.

**Weaknesses:**

1. Some conclusions seem not consistent across settings or need more verification: (1) 3 efficient methods improve safety for QwQ-32B, making me wonder if the effectiveness of efficiency methods on LRM safety highly depends on the model. The authors may need to verify it on more LRMs like Qwen3, GLM-4. Or reporting the average across all models and all datasets with a confidence score may also help better support the conclusion. (2) The correlation numbers in Figure 4 should be reported with confidence/significance metrics like p-value.
2. Some analysis could be more precise and detailed. For instance, "Efficiency methods often counterintuitively improve the safety of LRMs, but their effects depend strongly on the model and efficiency LLM technique." -- It's better to analyze quantitatively which methods improve safety, and which do not, and analyze the reason why there are differences between methods.
3. Presentation is poor in general: Figures 2 and 3 are far from the text that refers to them. Figure 6 is not referred to in the text. There is no figure or table comparison for visualization for the 2nd conclusion in section 4.3 (line 460).

**Questions:**

1. Are there works discussing the balance of efficiency and safety of LLMs before?

---

### Official Review · Reviewer_upKh · 2025-10-31

**Soundness:** 2
**Presentation:** 2
**Contribution:** 2
**Rating:** 2
**Confidence:** 4

**Summary:**

The paper investigates the effects of efficiency techniques: quantization (GPTQ, AWQ), pruning (SparseGPT, Wanda), and KV Cache compression (SnapKV, PyramidKV) on the safety, reasoning performance, and memory efficiency of three LRMs (QwQ-32B, R1D-Qwen-32B, R1D-Llama-8B). Using two jailbreak benchmarks (H-COT and Mousetrap) for safety and two reasoning benchmarks (MATH-500 and AIME) for performance, the authors find that these techniques often appear to improve safety (lower harmfulness ratings) but argue this is superficial, stemming from degraded reasoning capabilities leading to "attempted but failed" malicious responses. They also evaluate combinations of techniques, recommending quantization with pruning as optimal.

**Strengths:**

1. Timely topic (efficiency–safety tradeoffs for LRMs). The focus on LRMs is a reasonable extension, given their growing adoption and unique vulnerabilities.

2. Clear experimental story that “safety gains” can be superficial (failures rather than robust refusals).

3. Useful, practical baseline comparisons (GPTQ/AWQ, SparseGPT/Wanda, SnapKV/PyramidKV).

**Weaknesses:**

1. Benchmark breadth and construct validity (safety ≠ jailbreak only).
The work operationalizes “safety” solely as jailbreak harmfulness on two black-box suites with only 50 prompts each, scored by GPT-4o. This is a narrow slice of safety and a small N, which risks over-interpreting correlations. Broader trust/safety dimensions (toxicity, bias/fairness, honesty, privacy/PII leakage, impersonation, etc.) are not evaluated.
- context: The ICML’24 paper Decoding Compressed Trust [1] evaluates compression across eight trustworthiness dimensions and finds, e.g., moderate-bit quantization often preserves (sometimes improves) trustworthiness while pruning can hurt it, an important comparative frame that this submission does not engage with.

2. Insufficient analysis and lack of novel contribution. The paper primarily presents an observational study. Its main conclusion, that apparent safety gains are a byproduct of degraded reasoning ability, is an important data point, but it is not a particularly surprising insight. The work stops short of providing a deeper analysis or proposing novel solutions.

3. Statistical reporting is weak.
Results are point estimates only; no error bars, confidence intervals, or seed variation. With such small test sets and sampling-sensitive decoding, uncertainty is essential.

-----
[1] Hong et al. Decoding Compressed Trust: Scrutinizing the Trustworthiness of Efficient LLMs Under Compression. ICML 2024

**Questions:**

1. Could you explain the divergent conclusions from "Decoding Compressed Trust"? Does the chain-of-thought in LRMs make them more sensitive to compression than general LLMs?

---

### Official Review · Reviewer_zhFL · 2025-10-31

**Soundness:** 3
**Presentation:** 2
**Contribution:** 2
**Rating:** 2
**Confidence:** 3

**Summary:**

This paper studies how different efficiency-oriented techniques, including quantization, pruning, and KV-cache compression, affect both the reasoning performance and safety alignment of large reasoning models (LRMs). Main results suggest there is no universal impact with efficient techniques but a common tendency to preserve or even improve safety for LRMs. The authors conclude that many apparent “safety improvements” stem from reduced capability, i.e., attempted-but-failed behavior, rather than genuine alignment gains. They also explore different combinations of these methods and find that the Quant+Prune combination yields the best balance between reasoning and safety.

**Strengths:**

* It is critical to study how the safety performance is impacted by techniques of efficient LRMs. The study examines three key efficiency techniques and their combinations, offering an empirical picture across multiple models.
* The use of "attempted-but-failed" ratios highlights the difference between apparent and actual safety, which is a valuable observation.
* The results provide some useful insights for practical deployment that combined Quant+Prune strategies achieve better safety-reasoning tradeoffs.

**Weaknesses:**

* the contribution remains limited because it provides little mechanistic explanation for why different efficiency methods influence safety in distinct ways. A deeper analysis of internal model behaviors, like attention degradation, feature suppresion, would yield more meaningful insight beyond surface-level correlations.
* The paper’s interpretation of safety performance may be misleading. As shown by the “attempted-but-failed” behaviors, weaker or more aggressively compressed models can appear “safer” simply because they fail to generate responses. This makes the results in Section 4.1 less informative, as the harmfulness metric alone does not directly measure a model’s safety awareness or intention. If efficiency techniques merely reduce capability without improving alignment, the reported safety gains may not hold once capability is restored. Maybe the analysis should explicitly include compliance and refusal rates to distinguish genuine safety alignment from degraded ability. This could go a step further beyond harmful content but focusing on the intention to comply with malicious prompts.
* Previous studies [1] have sought to study how techniques like quantization affect LLM safety performance. In fact, [1] finds that quantization techniques uniformly degrade the safety of LLMs, which reaches the opposite conclusion. However, the authors have not included these work into discussion and comparison, leaving an important gap in related works. For instance, are there any unique features or characteristics in LRMs from LLMs that make a different conclusion ?
* The reported conclusions are inconsistent across models and methods, yet the experiments only cover DS and QwQ. Testing on a broader set of LRMs would make the findings more convincing and generalizable.

[1] Assessing Safety Risks and Quantization-aware Safety Patching for Quantized Large Language Models, ICML2025

**Questions:**

See weaknesses.

---

### Meta-Review · Area_Chair_6CQh · 2025-12-30

**Summary:**

This paper studied the safety-efficiency trade-off of large reasoning models. It analyzed different efficiency techniques like quantization, pruning and KV cache compression. The paper found that efficiency methods improve model safety but the improvement is due to the reduced reasoning capability.

The reviewers found that the paper focused on a timely and important topic of safety-efficiency trade-off. The paper provided a comprehensive analysis. However, there are multiple concerns about this paper. The contributions are limited in the sense that the paper provides little understanding on the analysis. The findings in the paper are not significant. And there are many related works in this field.

**Reviewer Concerns:**

The authors did not provide the rebuttal. Therefore, I believe the reviewer concerns are not addressed.

**Reviewer Scores:**

As the authors did not provide the rebuttal, the reviewer scores would not change.

---

### Decision · Program_Chairs · 2026-01-26

Reject